# Potential Regional Pollination Services of *Spodoptera litura* (Lepidoptera: Noctuidae) Migrants as Evidenced by the Identification of Attached Pollen

**DOI:** 10.3390/plants13243467

**Published:** 2024-12-11

**Authors:** Xiaokang Li, Huiru Jia, Dazhong Liu, Xianyong Zhou, Kongming Wu

**Affiliations:** 1College of Plant Protection, Shenyang Agricultural University, Shenyang 110866, China; lixiaokang2016@163.com; 2State Key Laboratory for Biology of Plant Diseases and Insect Pests, Institute of Plant Protection, Chinese Academy of Agricultural Sciences, Beijing 100193, China; 3Xianghu Laboratory, Hangzhou 311231, China; jhuiru@163.com (H.J.); liudazhong94@163.com (D.L.); zhouxy160721@163.com (X.Z.)

**Keywords:** *Spodoptera litura*, moth, pollen, pollination, migration

## Abstract

Many species of noctuid moths exhibit long-distance migratory behavior and have an important pollination service function in terrestrial ecosystems. *Spodoptera litura* (Fabricius) is a globally distributed insect; however, its role in pollination remains underexplored. In this study, the feeding preferences and inter regional pollination of *S. litura* adults were explored. We conducted pollen analysis on 1253 *S. litura* migrants captured from 2018 to 2021 on Beihuangcheng Island in the Bohai Strait of China, which is located in the East Asian insect migration path. The results show that an average of 51.1% of *S. litura* migrants carry plant pollen each year, and the carrying rate shows fluctuations based on sex, year, and season. By combining morphological identification and DNA barcoding, pollen species were identified from 40 species of plants, representing 21 families and 26 genera, mainly from angiosperms of Dicotyledoneae, with Asteraceae, Apocynaceae, and Amaranthaceae being the dominant taxa. The geographical distribution range of *Chrysanthemum zawadskii* and *Adenophora trachelioides* and a migration trajectory simulation analysis indicate that *S. litura* predominantly migrate from Liaoning Province in Northeast China to North China over the Bohai Sea in autumn. These findings indicate the potential pollination activities of *S. litura* in North China and Northeast China, enriching our understanding of the interaction between *S. litura* and the plants it pollinates.

## 1. Introduction

The relationship between plants and pollinators represents a critical ecological interaction within terrestrial ecosystems, with approximately 87.5% of angiosperms in nature relying on animal pollination for sexual reproduction [1]. The pollination behavior of insects promotes cross-pollination and maintains the genetic diversity of plants, which is important for the dynamic balance and relative stability of ecosystems [2,3]. In addition, animal pollination, which is dominated by insect pollination, is essential for improving crop yields and quality, as well as maintaining agricultural productivity [2,4]. Pollinating insects exhibit great diversity, mainly belonging to Hymenoptera, Diptera, Coleoptera, Lepidoptera, etc. [5]. The global decline in the diversity and abundance of pollinators is becoming evident, and this decline is due to multiple factors such as climate change [6], widespread pesticides use [7,8], changes of land use types [9,10], light pollution [11], and environmental pollution [12,13].

Existing research predominantly focuses on diurnal pollinating insects, represented by bees, significantly neglecting the study of nocturnal pollinators. The understanding of the scale of nocturnal insect pollination systems therefore remains insufficient [11,14]. However, nocturnal pollinators represent a crucial component of global biodiversity, as they provide essential pollination ecological services for both crops and wild plants [14,15]. In the context of the global decline in pollinating insects, the pollination ecological functions of some nocturnal insects urgently require accurate recognition and assessment. Attracted by the morphology, color, and odor of plant flowers, moths carry and transmit pollen on their body surfaces (proboscis, antenna, thorax, abdomen, leg, etc.) when visiting these flowers [11,15,16]. Compared with diurnal pollinators (bees, butterflies, etc.), the pollen transport network of moths is larger and more complex [15,17]. The pollination networks of moths and diurnal insects complement each other and help to maintain the diversity of plants.

Most long-distance migratory noctuid insects feed on nectar to supplement their nutrition during migration to complete normal flight and reproductive activities [18,19]. Moreover, the distribution and abundance of nectar-producing plants along migration routes also affect the completion of the long-distance migration of noctuid moths and their population outbreaks [20]. Many noctuid moth migrants, such as *Agrotis ipsilon* [21], *Mythimna seperata* [22], *Mamestra brassicae* [23], *Spodoptera exigua* [24], etc., carry pollen from various plants, demonstrating the pollination ecological service role of moth migration within the migration region. Furthermore, according to the geographical distribution area of pollen source plants visited by migrating insects, the migration sources and routes of insect migration can be inferred. *Vanessa cardui* butterflies captured in South America (French Guiana) were found to have carried pollen from plants endemic to the Sahel, directly confirming the migration of the butterfly across the Atlantic Ocean [25]. By correlating the trapping sites of insects with the geographical distribution of pollen source plants, the long-distance migration behavior of various noctuids has been confirmed [18,26]. Migration trajectory simulation represents an effective method for identifying the sources and landing areas of insect migration. The predictive results can accurately reflect the spatiotemporal dynamics of insect movements and can be used to predict exotic migratory pests.

The influence of small- to medium-scale low-level jet streams can directly influence the flight behavior of insects, which makes it possible to infer the flight trajectories of migratory insects using numerical meteorological background field models [27,28]. The hybrid single-particle Lagrangian integrated trajectory model (HYSPLIT) was specifically designed for analyzing the three-dimensional trajectories of particle spatial diffusion and deposition [29]. Meanwhile, the HYSPLIT model has also been extensively used in analyzing and forecasting the migration sources, flight paths, and landing zones of various migratory insects [30,31,32,33]. The consistency between the field occurrence dynamics of insects and the trajectory simulation results in the prediction area has indicated the accuracy of the HYSPLIT model for insect migration trajectory simulation [34,35].

*Spodoptera litura* is a worldwide omnivorous pest. Its larvae can feed on nearly three hundred kinds of plants, including food crops, economic crops, fruit trees, vegetables, flowers, etc. [36]. Various research methods, such as marine ship insects capture [37], migration population monitoring [38], field population dynamics monitoring [34], trajectory simulation [30,34], and genetic diversity analysis of different geographical populations [39,40], have shown that *S. litura* can migrate seasonally in subtropical and temperate regions of Asia.

The seasonal migration of *S. litura* across the Bohai Sea has been reported, mainly occurring from August to October each year [38]. In this study, based on monitoring historical data from April to October for the period 2018–2021, we obtained individuals of the *S. litura* migrants on Beihuangcheng Island (BHC) using searchlight traps and discovered pollen on their body surfaces using optical microscopy. Additionally, through pollen micro-morphological observation and DNA barcoding analysis, we identified the species of pollen carried by *S. litura* migrants. Based on the species and geographical distribution of pollen source plants, the cross regional pollination service function of adult *S. litura* migrants was analyzed.

## 2. Results

### 2.1. Variation Characteristics of Pollen Adherence Rate of Migrating S. litura

A total of 1253 *S. litura* individuals was collected for pollen analysis, and the number of individuals carrying pollen were recorded. The migrating moths carried pollen from multiple plants species, of which 83.8% adhered by pollen grains from one single plant species, 14.7% carried pollen from two plant species, while the remainder of individuals harbored pollen grains of multiple species. There was no significant difference in the annual pollen carrying rate from 2018 to 2021 (*χ*^2^ = 2.828, *df* = 3, *p* = 0.419), with the highest pollen adherence recorded in 2018 (Table 1).

The monitoring data obtained using the light traps showed that *S. litura* mainly migrated across the Bohai Sea from August to October, with fewer migrations in June and July. In the months of June and July from 2018 to 2021, the number of *S. litura* individuals carrying pollen varied greatly. In June 2020, pollen adhered to 16.7% (1/6) migrants; in June 2021, all moths carried pollen grains (5/5), and in July, the pollen carrying rate of *S. litura* was 38.9% (14/36) (Figure 1). However, from August to October during the period 2018–2021, the frequency of moths with adhering pollen did not differ significantly (*F*_2, 9_ = 3.886, *p* = 0.061), with the maximum value in October (67.3 ± 7.0%), followed by August (49.4 ± 10.3%) and September (38.0 ± 2.5%).

During the period from 2019 to 2020, there was no significant sex difference in the pollen carrying rates of *S. litura* migrants in a given year (2019: *χ*^2^ = 2.57, *df* = 1, *p* = 0.109; 2020: *χ*^2^ = 0.682, *df* = 1, *p* = 0.409) (Figure 2). However, the pollen carrying rate of *S. litura* females (56.8%) was significantly higher than that of males (44.6%) in 2021 (*χ*^2^ = 4.291, *df* = 1, *p* = 0.038) (Figure 2). Moreover, the overall analysis of the microscopic examination data showed no significant difference in the pollen carrying rate of male (51.3 ± 3.4%) and female moths (51.7 ± 4.4%) from 2019 to 2021 (*t =* 0.089, *df =* 4, *p =* 0.934) (Figure 2).

### 2.2. Pollen Source Plants Visited by Migrating S. litura

A total of 392 SEM images of pollen grains were obtained. Based on the pollen morphology and DNA metabarcoding, a total of 40 morpho-types of pollen grains across 26 genera and 21 families was recorded (Table 2). In particular, 16 morpho-types of pollen grains were identified to species level as follows: *Cynanchum rostellatum* (Turcz.) Liede & Khanum, *C. chinense* R. Br., *Vincetoxicum atratum* (Bunge) Morren & Decne., *Albizia julibrissin* Durazz., *Lycium chinense* Mill., *Atractylodes lancea* (Thunb.) DC., *Artemisia argyi* H. Lév. & Vaniot, *Chrysanthemum lavandulifolium* (Fisch. ex Trautv.) Makino, *Ch. Zawadskii* Herbich, *Adenophora trachelioides* Maxim., *Humulus scandens* (Lour.) Merr., *Eleusine indica* (L.) Gaertn., *Tamarix chinensis* Lour., *Flueggea suffruticosa* (Pall.) Baill., *Cuscuta japonica* Choisy, *Suaeda glauca* (Bunge) Bunge. Most morpho-types of the pollen grains were identified to family or genus level, and the micrographs are shown in Figure 3. Asteraceae, Apocynaceae, and Amaranthaceae plants dominated the pollen source plant species visited by *S. litura* migrants (Table 2 and Appendix A).

By analyzing the pollen grains carried by *S. litura* migrants from June to October, we found that the proportion of different families represented by adhering pollen varies. In June–July, the carried pollen species were derived from Simaroubaceae (40.0%), Pinaceae (20.0%), Vitaceae (20.0%), and Poaceae (20.0%). As the season progressed, pollen gains from Asteraceae plants became commonly recorded, with 40.9% (August), 39.3% (September), and 80.5% (October). Collectively, pollen grains from Asteraceae (63.5%), Chenopodiaceae (9.9%), and Solanaceae (4.2%) plants were recorded more frequently than others (Table 3).

Based on the taxonomic characteristics, comparative analyses of pollen source plants of the cross-sea migrating *S. litura* from 2018 to 2021 were conducted. The results show that the percentage of *S. litura* migrants visiting angiosperms was significantly higher than that of gymnosperms (*χ*^2^ = 19.174, *df* = 1, *p* < 0.001). The percentage of dicotyledonous pollen source plants (91.3%) was significantly higher than that of monocotyledonous plants (8.7%) (*χ*^2^ = 15.696, *df* = 1, *p* < 0.001). Captured *S. litura* migrants visited herbaceous plants more often than woody plants (*χ*^2^ = 9.783, *df* = 1, *p =* 0.002) (Figure 4). The taxonomic information regarding pollen source plants visited by migrating *S. litura* individuals can be found in Appendix A.

### 2.3. Possible Migration Source Areas of S. litura Across the Bohai Sea in Autumn

From 2018 to 2021, among the 16 pollen source plants isolated and identified to species level, most exhibited a relatively broad geographical distribution, with a limited indication of migration source areas (Appendix A). However, during the migration period from August to October, *Ch. zawadskii* and *Ad. trachelioides* pollen grains were detected on *S. litura* migrants. These two plants are mainly distributed in Northeast and North China (Figure 5a,b), indicating that *S. litura* adults trapped in BHC may have migrated from these areas.

The dates of *S. litura* individuals carrying these two types of pollen are listed in Appendix A. The trajectory analysis of these dates shows that the source areas of *S. litura* autumn migrants mainly include Liaoning Province, Inner Mongolia, Jilin Province, and Shandong Province, as well as parts of North Korea and South Korea. The backward trajectory simulation results for moths carrying *Ch. zawadskii* pollen grains show that the endpoints were distributed in Liaoning Province (84.43%), North Korea (11.75%), Jilin Province (1.53%), South Korea (1.41%), Inner Mongolia (0.61%), Shandong Province (0.19%), Tianjin (0.04%), and Heilongjiang Province (0.04%) (Figure 5c). The backward trajectories of migrants carrying *Ad. trachelioides* pollen grains show that the endpoints were distributed in Liaoning Province (61.85%), Shandong Province (38.01%), and Tianjin (0.14%) (Figure 5d). The endpoints of the backward trajectories were selected and retained based on the geographical distributions of *Ch. zawadskii* and *Ad. trachelioides* (Figure 5e,f). Specifically, the proportion of trajectory endpoints within the distribution area of *Ch. zawadskii* in Liaoning Province was 95.19%, and in Inner Mongolia it was 4.81% (Figure 5g). In the distribution area of *Ad. trachelioides*, the trajectory endpoints accounted for 92.57% in Liaoning Province and 7.43% in Shandong Province (Figure 5h). In summary, Liaoning Province was the main source of fall migration of *S. litura* across the Bohai Sea, with a few migrating individuals coming from Shandong Province and Inner Mongolia.

## 3. Discussion

BHC is located in the depth of the Bohai Strait, about 40 km away from the land. No *S. litura* larvae were found on BHC during the monitoring period. Upon combining the data from the monitoring light traps (unpublished data) and field larvae investigations, it was indicated that *S. litura* trapped on BHC originate from migratory populations rather than local breeding. Many migratory insects, including noctuid moths, carry pollen grains from various plants and play an important role in pollination during their cross-regional migration [15,64]. Based on the morphology of pollen and DNA barcoding, we identified 40 morpho-types of pollen species belonging to 26 genera across 21 families from the surface of *S. litura* migrants. This suggests that *S. litura* had visited at least the listed pollen source plants during its long-distance migration. This was the identification result of the migration population for only one monitoring site, showing the diversity of pollen source plants that *S. litura* adults feed on and the potential pollination role of *S. litura* adults.

In this study, pollen morphology identification and DNA barcoding were combined to identify and analyze pollen grains carried by *S. litura* migrants. The typical method for identifying pollen species relies on their microscopic morphological characteristics, including shape, symmetry, polarity, ornamentation types, apertures, etc. [65]. However, morphological identification requires specialized palynological knowledge and experience, and the results of such identifications may lack accuracy, presenting significant limitations [65,66]. DNA barcoding has gained increasing application in plant taxonomy due to its operability and accuracy [67,68]. In this study, *rbcL*, *psbA-trnH*, and ITS were selected for plant barcode amplifications. A total of 40 pollen source plants species was identified from the body surfaces of *S. litura* migrants, indicating the effectiveness of the combination of pollen morphological characteristics with pollen DNA barcodes in identifying pollen species carried by *S. litura* individuals. However, among the 40 morpho-types of pollen grains, 17 types were only identified at the genus or family level, and the taxonomic status of 8 morpho-types of pollen grains was unknown, indicating the limitations of the combination of the two methods in pollen identification. Because of its high throughput and sensitivity, the metabarcoding technique is increasingly being applied in the identification of insect-borne pollen, identifying links in insect pollination networks [69,70], and the analysis of insect migration sources [25,71,72]. In our future research, we hope to improve the efficiency of the detection and identification of pollen carried by noctuid moths using the metabarcoding technique to gain a deeper understanding of noctuid moths’ pollination function and migration routes.

There was no significant monthly difference in the pollen carrying rate of migrating *S. litura* adults, while the frequency of pollen adherence of spring-migrating noctuid moths such as *Ag. ipsilon* [21], *Ag. segetum* [26], and *M. brassicae* [23] migrants was higher than that of summer- and fall-migrating individuals. The flower-visiting frequency of insects is affected by various factors, such as plant phenology [73] and floral characteristics (color, shape, odor, nectar content, etc.). Unlike the migration of other noctuid insects in spring, summer, and autumn, the migration of *S. litura* mainly occurs in autumn, with a concentrated period of occurrence. The flowering characteristics of plants in autumn may be the main reason for the absence of monthly differences in the pollen carrying rates of *S. litura* migrants. Our results demonstrate that male and female *S. litura* migrants do not differ significantly in the frequency of pollen adherence; similar findings have been recorded for *Ag. ipsilon* [21], *Mythimna separata* [22], *Ag. segetum* [26], and *M. brassicae* [23]. Intraspecific sex differences in pollination behavior, such as flower visits frequency, types of flowers, and pollen transfer, have been widely observed [74,75,76]. The pollination status of plants can also affect the oviposition selection of insects [77,78,79]. The trade-off between insect flight and reproduction can also affect insect pollination behavior, and the increased allocation of resources for flight may enhance the efficiency of lepidopteran pollinators, subsequently improving the adaptability of plants [80]. Therefore, it may be necessary to explore the reasons for the sex differences of many migratory moths, from multiple aspects such as individual nutritional requirements, oviposition preference, and trade-offs between flight and reproduction.

In June–July, the pollen grains of Simaroubaceae, Pinaceae, Vitaceae, and Poaceae were identified from migrating *S. litura* individuals. The pollen quantity of pollen grains in each family is relatively balanced, which may be due to the small number of captured and tested migrants. Starting from August, pollen from Asteraceae plants became the main type of pollen carried by migratory individuals of *S. litura*. Even in October, the pollen of Asteraceae plants reached 80.5%. This may be related to the fact that the identified *Chrysanthemum*, *Chenopodium*, and *Artemisia* plants mostly bloom in autumn and have a large amount of pollen grains [56]. A similar phenomenon has also been observed in previous studies on *M. brassicae* [23] and *A. segetum* [26]. In addition to the preference of moths to visit flowers themselves, the richness of flowering plant species within the habitat is also a major factor affecting insects feeding choices. The migrating individuals of *S. litura* mainly visited the flowers of angiosperms and only visited one kind of Pinaceae plant, which is consistent with the feeding preferences of other adult noctuid moths, such as *Ag. ipsilon* [21], *Ag. segetum* [26], *M. brassicae* [23], and *S. exigua* [24]. The proportion of herbaceous pollen source plants visited by *S. litura* was significantly higher than that of woody plants, and was similar to that for *Ag. segetum* [26]. Previous studies have shown that the feeding preferences of adult insects can affect population dynamics. For example, supplementing nutrition can affect the population fitness of *Helicoverpa armigera* [81]. After feeding on pollen or nectar from different plants, there are interspecific differences in the population fitness changes for *M. Separata*, *M. loreyi*, *Athetis lepigone*, and *Hadula trifolii* [82]. Meanwhile, studies have shown that many Lepidoptera insects obtain nectar from the host plants of their larvae [83]. Hence, studying the host plant preference of migrating *S. litura* adults can provide a foundation for the subsequent research on the effects of adult host-plant preferences and host distribution on its population dynamics.

According to pollen morphological characteristics and pollination types, the pollen source plants visited by *S. litura*, such as *C. rostellatum*, *C. chinense*, *V. atratum*, *A. julibrissin*, *L. chinense*, *At. lancea*, *Ch. lavandulifolium*, *Ch. zawadskii*, *Ad. trachelioides*, *T. chinensis*, *F. suffruticosa*, and *Cu. japonica*, are typical entomophilous plants, whose pollen grains present an oblate shape with rough surface ornamentation [84,85]. In addition, entomophilous flowers can emit aromatic odors and have nectar glands that secrete nectar, attracting insects to feed on them and perform pollination [86,87]. Long-distance migration of insects is a significant energy-consuming process. Insects need to feed on nectar plants to meet energy needs during long-distance migration. For *S. litura* individuals, long-distance migration is also a significant energy-consuming process, and they also need to feed on nectar to meet their energy requirements. Particularly, *S. litura* migrants carry pollen from *Ar. argyi*, *S. glauca*, *Amaranthus*, *Chenopodium*, *Ambrosia*, *Ailanthus*, Pinaceae, and other anemophilous plants. The pollen grains of these anemophilous plants are small and lightweight, with smooth surface ornamentation, and some even have air sacs [84,85]. Anemophilous flowers are small and not bright, mostly odorless, do not have nectaries, and are not attractive to noctuid moths [86,88]. However, in addition to *S. litura*, many noctuids transport significant amounts of anemophilous plant pollen grains during their migration [18,20,21,22,23,26,81]. Pollen grains of anemophilous plants may settle on the stigmas of entomophilous plants via airflow [89,90]. Deposited pollen from anemophilous plants may adhere to the body surfaces of *S. litura* while they are visiting these entomophilous flowers. The pollination of many entomophilous plants is influenced by biotic and abiotic factors. Under certain conditions, insects also visit the flowers of anemophilous plants and facilitate pollination, a phenomenon known as ambophily [91,92,93]. The active visitation and contact of *S. litura* with these anemophilous flowers may be another reason for the detection of their pollen grains.

Pollen, as a biomarker, is widely used to explore the migration routes and patterns of migratory insects. By identifying and analyzing pollen grains carried by *V. cardui*, its long-distance migration between the Middle East and Europe, as well as between the Middle East and South America, has been clarified [25,71,72]. Spring migratory moths such as *Ag. ipsilon* [21], *Ag. segetum* [26], *Ha. trifolii* [18], and *M. brassicae* [23] captured on BHC carried pollen grains of plants distributed in central, southwestern or southern China, indicating the source areas of these spring migratory individuals. The pollen source plants visited by *S. litura* are widely distributed in China. Among them, only *Ch. zawadskii* and *Ad. trachelioides* are mainly distributed in northern and northeastern China, which limits the identification of the source areas of *S. litura*. BHC is located within the East Asian Monsoon Region, with strong monsoon winds in summer and winter, and northerly winds prevailing from August [94,95]. Combined with the distribution areas of the pollen source plants, the trajectory simulation further indicates that *S. litura* mainly took off from Liaoning Province in autumn and migrated southward over the Bohai Sea. The autonomous flight behavior (flight speed, orientation behavior, etc.) of insects has a significant impact on the simulated migration trajectories [96,97]. Our trajectory simulation was performed with the addition of flight parameters such as insect heading offset and flight speed, achieving accurate simulation results. These flight parameters were extracted from the X-band scanning entomological radar monitoring data of typical migration events of *S. litura* [38]. Insect species cannot be directly extracted from X-band scanning entomological radar monitoring data [98]. However, because of the great migratory biomass of *S. litura* in typical events, the biological parameters were obtained using statistical analysis. Importantly, the effects of environmental factors and meteorological conditions on insect migration have spatial and temporal heterogeneity. It is important to accurately obtain the actual flight parameters of insects and construct a more accurate simulation model.

## 4. Materials and Methods

### 4.1. Collection of Migratory S. litura Individuals

Migratory *S. litura* individuals were obtained by the self-designed high-altitude searchlight trap at the Changdao Experimental Base of the Institute of Plant Protection, Chinese Academy of Agricultural Sciences (38.3869° N, 120.9092° E), located on Beihuangcheng Island (Figure 6a). The trap mainly consists of an iron frame, a lampshade, a 1000 W metal halide lamp, and an insect collection bag (Figure 6b,c), which was placed on the ground without obstacles around it. According to the climate conditions and historical monitoring data, we conducted monitoring of the migration of *S. litura* from April to October each year. During the monitoring period, the light trap was turned on before sunset on the previous day and turned off after sunrise on the next morning (the working state of the light trap is shown in Figure 6d). In the case of extreme rainfall or power outages, the light trap was not turned on and the monitoring work was suspended. After turning off the light trap, the insect collecting bag was retrieved and placed in a −20 °C freezer for several hours. Subsequently, *S. litura* migrants were picked out of the mixed insects and each moth was stored in a 2 mL tube; the capture date and sex of moths were recorded. Next, *S. litura* migrants were frozen at −20 °C until the microscopic examination was carried out.

### 4.2. Acquisition and Microscopic Observation of Pollen Grains

The head of each *S. litura* individual was removed using dissecting forceps and placed on a glass slide. We sequentially examined the antenna, compound eyes, proboscis, and labial palpus for the presence of pollen under a stereomicroscope (SZX16, Olympus, Tokyo, Japan). Whether the individuals of *S. litura* carried pollen grains and the amount of carried pollen grains were recorded. Glass capillaries (hard neutral glass, inner diameter 0.5 mm, length 100 mm) were pulled apart from the middle using a micropipette puller (P-1000, Sutter, Alpharetta, GA, USA), and then fixed in pipet tips. The setting parameters of the micropipette puller are shown in Appendix A. Pollen grains were collected and transferred to tapes (carbon tape attached to aluminum stubs, 8 mm wide, NISSHIN, Tokyo, Japan) attached to the scanning electron microscope (SEM) pin mount (50 mm plane, HOGO, Kunshan, China) using self-made glass needles. The pollen grains with extremely similar morphology under the stereomicroscope were randomly selected and attached to the SEM pin mount. Subsequently, pollen grains were sprayed with gold palladium using an ion sputter coater (IB-5, EIKO, Tokyo, Japan) and then observed under SEMs (S4800/Regulus 8100, HITACHI, Tokyo, Japan). To prevent cross-contamination, a single pollen grain was selected with a single needle. Slides and tweezers were wiped with 75% alcohol wipes before each microscopic examination.

### 4.3. DNA Extraction and PCR of Pollen Grains

Single pollen DNA was obtained using the alkaline lysis method. The lysis solution was mixed with 1M NaOH (Sinopharm, Shanghai, China), 20% Tween-20 (Amresco, Framingham, MA, USA), and ddH_2_O at a ratio of 1:1:8. A single pollen grain photographed using SEM was selected and transferred into a 0.2 mL PCR tube containing 5 μL lysis solution using a glass needle. The PCR tube was centrifuged briefly and then heated at 95 °C for 17.5 min. Once the procedure was completed, 5 μL TE (Tris-EDTA) buffer (Coolaber, Beijing, China) was added to the above lysate to form the PCR template solution.

*rbc*L and *psbA*-*trnH* genes in chloroplast DNA, and ITS in nuclear DNA, were selected for pollen identification. The sequence of the primers is shown in Appendix A. The conditions for PCR were set according to a previous study [26]. The PCR products were purified using a Gel Extraction Kit (Axygen, Corning, NY, USA), and ligated to the Trans1-T1 vector (TransGen, Beijing, China). Subsequently, positive colonies were screened and sequenced.

### 4.4. Pollen Species Identification

The obtained pollen electron microscope images were compared with illustrations in professional books and literature on pollen morphological features [41,42,43,44,45,46,47,48,49,50,51,52,53,54,55,56,57,58,59,60,61,62,63]. Based on their morphological characteristics, these images were classified. Region of similarity comparisons of pollen grain DNA sequences obtained using the Sanger method were performed using the BLAST tool on the NCBI website (https://blast.ncbi.nlm.nih.gov/Blast.cgi, accessed on 10 April 2023) [99,100]. The sequence alignment method evaluates the similarity of the sampled sequence based on the best hit of the query sequences. The Expect Value (E-value) refers to the expected number of random hits for a given alignment score, which was selected as the main index during the alignments. According to the classification of the five species with the highest scores in sequence alignment, the taxa of sampled pollen source plants were determined using previous research methods [21,101]. Based on the comparison of pollen morphology and DNA barcoding, the taxonomic status and species of pollen source plants were inferred. After the identification of pollen, the types of pollen source plants were classified to determine the feeding preferences of *S. litura* migrants.

### 4.5. Analysis of Migration Source Areas of S. litura Across the Bohai Sea

Based on the specific geographical distribution of pollen source plants and migration trajectory simulations, the source areas of *S. litura* undertaking across-sea migration were inferred. The geographical distribution of pollen source plants was retrieved from a professional botany website (https://www.iplant.cn/, accessed on 8 August 2024). Pollen source plants with specific distribution areas were screened, and their specific distribution areas indicated the source area of *S. litura* migrants.

The source areas of *S. litura* migrants were further confirmed using backward trajectory analysis. Trajectory simulations for dates on which *S. litura* individuals carrying indicative pollens were trapped were performed using an improved model based on HYSPLIT 5.3.0, which incorporated biological parameters such as insect heading offset and flight speed. The fifth generation of atmospheric reanalysis data (ERA5) with latitude-longitude grids at 0.25° × 0.25°, within the latitude-longitude ranges of 25–50° N and 105–130° E, was obtained from the website (https://cds.climate.copernicus.eu/, accessed on 5 September 2024) for trajectory simulation. The capture dates of individuals carrying indicative pollen grains were selected for trajectory simulation. BHC was set as the end location of the trajectory simulation. With reference to the operating time of the searchlight trap, trajectories were simulated every hour from 19:00 each night to 06:00 the next morning (Beijing time, UTC+8).

Trajectory simulation of *S. litura* flight heights were set as per a previous study [101]: 250 m, 500 m, 750 m, 1000 m, and 1250 m above mean sea level. Because of the strong flight ability of *S. litura*, a 30° insect heading offset (the angle between the insect head orientation and wind direction) and 2.5 m s^−1^ insect flight speed [38,102] were added for the trajectory simulation. The maximum simulated flight duration was set to 12 h. Simultaneously, the start locations of the backward trajectory simulation were screened, and only the locations that fell into the geographical distribution of specific pollen source plants represented the migration source areas of *S. litura* migrants.

### 4.6. Data Analysis

The rates of pollen adhering to *S. litura* migrants in different months were analyzed using one-way ANOVA, and the differences were identified using Tukey’s honestly significant difference test. Differences in the annual mean between *S. litura* male and female moths were analyzed using Student’s *t*-test. Sex differences in the annual mean frequencies of pollen deposits on *S. litura* moths and differences in the taxonomic status of pollen source plants were all analyzed using a Chi-squared test. All percentage data were logit transformed, and further analyses were carried out in SPSS 22.0 (IBM Corp, New York, NY, USA).

## 5. Conclusions

As pollinators of numerous plants, noctuid moths play an important ecological service function. In this study, we identified the species of pollen grains carried by *S. litura* migrants across the Bohai Sea. Upon combining morphology-based identification and DNA barcoding, 40 species of plants belonging to 26 genera in 21 families were identified, indicating significant nocturnal flower visitations of *S. litura*. Based on the geographical distribution areas of *Ch. zawadskii* and *Ad. trachelioides*, as well as trajectory simulation, *S. litura* mainly migrated from Liaoning Province in northeastern China to northern China in autumn. Our results demonstrate that flower visiting and pollen carrying behavior during the long-distance migration makes it possible for *S. litura* to engage in cross-regional pollination. Subsequent studies should aim to determine the importance of this moth species in pollination. Our results are helpful for understanding the trophic relationship and coevolution between *S. litura* and its host plants over large geographical scales.

## Figures and Tables

**Figure 1 plants-13-03467-f001:**
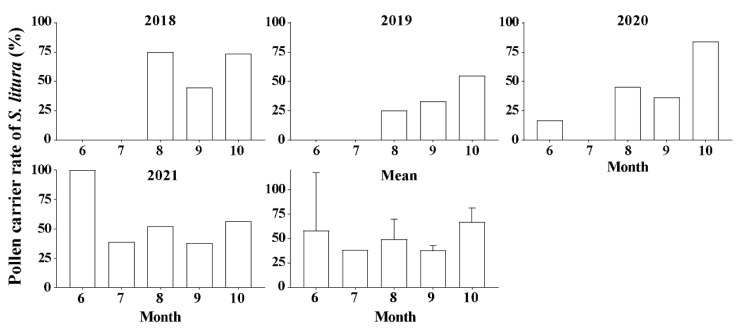
Pollen carrying rate of *S. litura* migrants on BHC in different months from 2018 to 2021. The mean data are presented as mean ± SE.

**Figure 2 plants-13-03467-f002:**
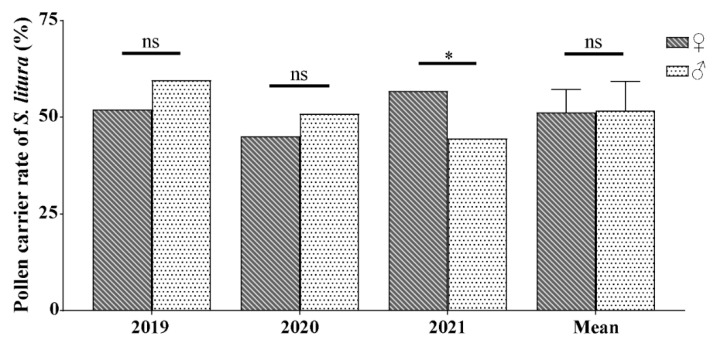
Variations in pollen carrying rates among male and female of *S. litura* migrants on BHC from 2019 to 2021. The mean data are presented as mean ± SE. The symbol * indicates a significant difference based on the *t*-test (*p* < 0.05), while ns indicates no significant difference between males and females.

**Figure 3 plants-13-03467-f003:**
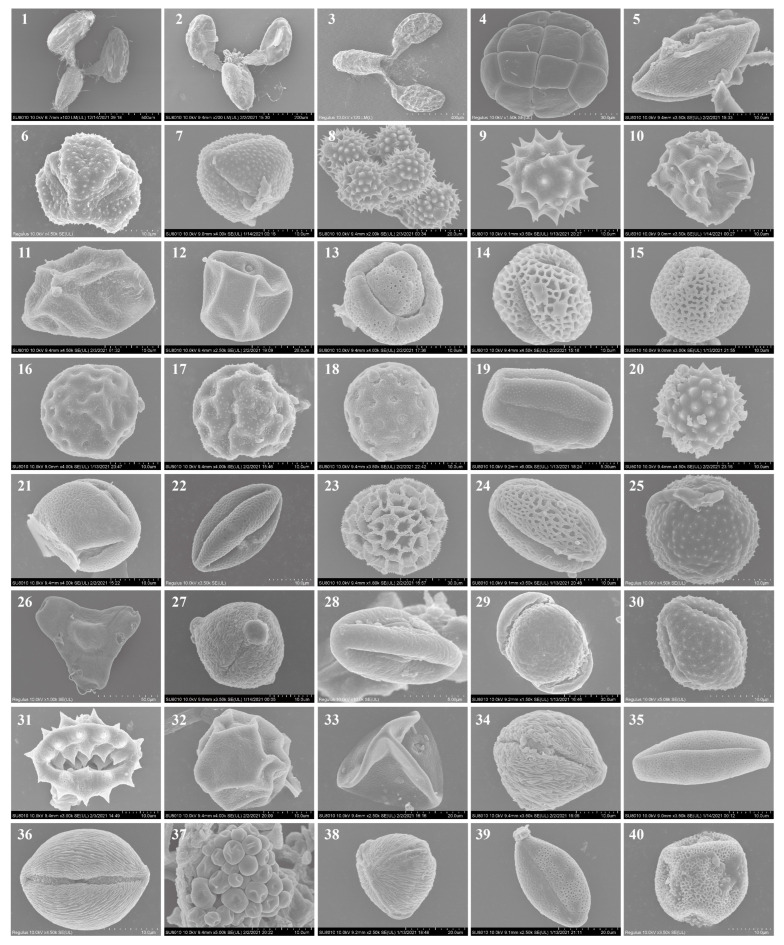
Microscopic images of the pollens carried by *S. litura* migrants from 2018 to 2021. 1. *Cynanchum rostellatum*, 2. *C. chinense*, 3. *Vincetoxicum atratum*, 4. *Albizia julibrissin*, 5. *Lycium chinense*, 6. *Atractylodes lancea*, 7. *Artemisia argyi*, 8. *Chrysanthemum lavandulifolium*, 9. *Ch. zawadskii*, 10. *Adenophora trachelioides*, 11. *Humulus scandens*, 12. *Eleusine indica*, 13. *Tamarix chinensis*, 14. *Flueggea suffruticosa*, 15. *Cuscuta japonica*, 16. *Suaeda glauca*, 17. *Amaranthus*, 18. *Chenopodium*, 19. *Rubia*, 20. *Ambrosia*, 21. *Vitis*, 22. *Fagopyrum*, 23. *Limonium*, 24. *Ailanthus*, 25. *Clematis*, 26. *Epilobium*, 27. *Orostachys*, 28. *Castanea*, 29. Pinaceae, 30–31. Asteraceae, 32–33. Poaceae, 34–40. Unknown.

**Figure 4 plants-13-03467-f004:**
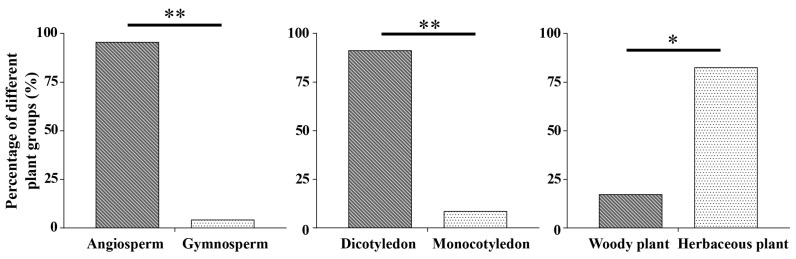
Classification of pollen source plants of *S. litura* migrants on BHC from 2018 to 2021. The symbol * indicates significant difference via Chi-squared test (*p* < 0.05), while the symbol ** indicates extremely significant difference (*p* < 0.001).

**Figure 5 plants-13-03467-f005:**
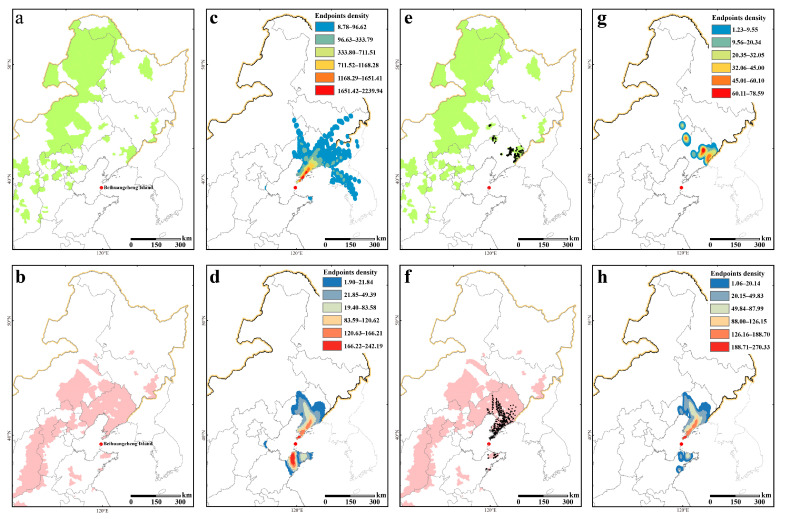
Migration source areas of *S. litura* across the Bohai Sea indicated by carried pollen and trajectory simulation. (**a**,**b**) Geographical distribution areas of *Ch. zawadskii* and *Ad. trachelioides*, respectively. (**c**,**d**) Effective endpoints for the trajectory simulation of *S. litura* carrying pollen of *Ch. zawadskii* and *Ad. trachelioides*. (**e**–**h**) Effective endpoints for the trajectory simulation of *S. litura* in the geographical distribution areas of *Ch. zawadskii* and *Ad. trachelioides*, respectively.

**Figure 6 plants-13-03467-f006:**
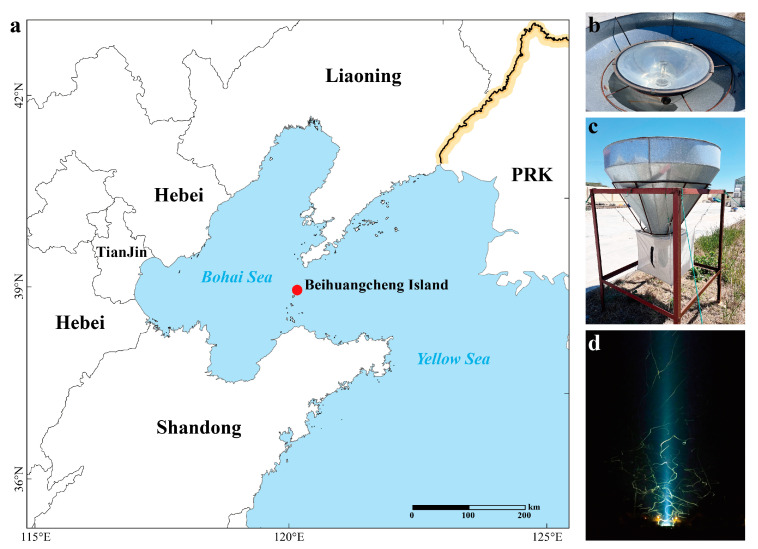
Schematic diagram of the migration monitoring site and high-altitude searchlight trap used to collect *S. litura* specimens: (**a**) Location of Beihuangcheng Island. (**b**,**c**) Images of the light trap. (**d**) Working state of the light trap.

**Table 1 plants-13-03467-t001:** Inter-annual pollen carrying rate of *S. litura*.

No./%	2018	2019	2020	2021	Mean ± SE
No. moths examined	86	666	199	302	313.2 ± 108.8
No. moths carrying pollen	50	325	96	149	155 ± 52.1
Pollen carrying rate (%)	58.1	48.8	48.2	49.3	51.1 ± 2.0%

**Table 2 plants-13-03467-t002:** Molecular and morphological identification of pollens carried by adult *S. litura* migrants on BHC from 2018 to 2021.

Num.	Identified Plants	Molecular Identification			References for Morphology	Pollination Type
		*rbc*L	ITS	*trn*H-*psb*A		
1	*Cynanchum rostellatum*	+, *C. rostellatum*	+, *C. rostellatum*	+, *Cynanchum* L.	[41]	Entomophily
2	*C. chinense*	+, *C. chinense*	+, *C. chinense*	+, *C. chinense*	[42]	Entomophily
3	*Vincetoxicum atratum*	+, *Cynanchum* L.	+, *Cynanchum* L.	-	[43]	Entomophily
4	*Albizia julibrissin*	+, *A. julibrissin*	+, *A. julibrissin*	-	[44]	Entomophily
5	*Lycium chinense*	+, *L. chinense*	-	-	[45]	Entomophily
6	*Atractylodes lancea*	-	+, *At. lancea*	-	[46]	Entomophily
7	*Artemisia argyi*	+, *Ar. argyi*	+, *Ar. argyi*	-	[47]	Anemophily
8	*Chrysanthemum lavandulifolium*	-	+, *Ch. lavandulifolium*	-	[48]	Entomophily
9	*Ch. zawadskii*	+, *Ch. zawadskii*	-	-	[48]	Entomophily
10	*Adenophora trachelioides*	-	+, *Ad. trachelioides*	-	[49]	Entomophily
11	*Humulus scandens*	-	+, *H. scandens*	-	[50]	Ambophily
12	*Eleusine indica*	-	+, *E. indica*	-	[51]	Ambophily
13	*Tamarix chinensis*	+, *T. chinensis*	-	-	[52]	Entomophily
14	*Flueggea suffruticosa*	-	+, *F. suffruticosa*	-	[53]	Entomophily
15	*Cuscuta japonica*	-	+, *Cu. japonica*	+, *Cu. japonica*	[54]	Entomophily
16	*Suaeda glauca*	+, *S. glauca*	+, *S. glauca*	+, *Suaeda* Forssk. ex J. F. Gmel.	[55]	Anemophily
17	*Amaranthus* L.	-	-	-	[56]	Anemophily
18	*Chenopodium* L.	-	-	-	[56]	Anemophily
19	*Rubia* L.	-	-	-	[56,57]	Entomophily
20	*Ambrosia* L.	-	-	-	[56,58]	Anemophily
21	*Vitis* L.	-	-	-	[59]	Ambophily
22	*Fagopyrum* Mill.	-	-	-	[60]	Entomophily
23	*Limonium* Mill.	-	-	-	[61]	Entomophily
24	*Ailanthus* Desf.	-	-	-	[44]	Ambophily
25	*Clematis* L.	-	-	-	[62]	Entomophily
26	*Epilobium* L.	-	-	-	[62]	Ambophily
27	*Orostachys* (DC.) Fisch.	-	-	-	[56]	Entomophily
28	*Castanea* Mill.	-	-	-	[63]	Ambophily
29	Pinaceae	+, *Pinus bungeana*	-	-	[44,62]	Anemophily
30	Asteraceae	-	-	-	[56,62]	Anemophily
31	Asteraceae	-	-	-	[56,62]	Entomophily
32	Poaceae	-	-	-	[51,62]	Anemophily
33	Poaceae	-	-	-	[51,62]	Anemophily
34–40	Unknown	-	-	-	-	

Note: “+” indicates successful sequence amplification and alignment; “-” indicates failed sequence amplification or alignment.

**Table 3 plants-13-03467-t003:** Relative percentage of different families represented by pollen grains adhering to *S. litura* migrants captured on BHC from 2018 to 2021.

Family	June–July	August	September	October	Overall
Simaroubaceae	40.0	-	-	-	0.6
Pinaceae	20.0	-	0.9	1.0	1.2
Vitaceae	20.0	-	-	-	0.3
Poaceae	20.0	4.5	3.7	3.5	3.9
Campanulaceae	-	22.7	-	1.5	3.9
Asteraceae	-	40.9	39.3	80.5	63.5
Leguminosae	-	4.5	-	-	0.3
Chenopodiaceae	-	9.1	9.3	10.5	9.9
Onagraceae	-	4.5	-	-	0.3
Euphorbiaceae	-	9.1	4.7	-	2.1
Polygonaceae	-	4.5	-	-	0.3
Ranunculaceae	-	-	6.5	-	2.1
Apocynaceae	-	-	4.7	0.5	1.8
Solanaceae	-	-	12.1	0.5	4.2
Cannabaceae	-	-	2.8	-	0.9
Plumbaginaceae	-	-	2.8	0.5	1.2
Crassulaceae	-	-	0.9	-	0.3
Rubiaceae	-	-	2.8	-	0.9
Convolvulaceae	-	-	4.7	-	1.5
Tamaricaceae	-	-	-	1.0	0.6
Fagaceae	-	-	-	0.5	0.3

## Data Availability

Data are contained within the article or Appendix A.

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
