# Peer review of "Potential Regional Pollination Services of Spodoptera litura (Lepidoptera: Noctuidae) Migrants as Evidenced by the Identification of Attached Pollen"

_plants, 2024, doi:10.3390/plants13243467_

Round 1
Reviewer 1 Report
Comments and Suggestions for Authors
The authors questioned the migratory routes of Spodoptera litura, analysing the pollen carried by migratory individuals. It is quite a standard method, but it is still interesting when applied to a new migratory species and area. They used a robust flying prediction model to extrapolate the moth's flying routes and cross-link those data with the botanical origin of the carried pollen to find the possible last stop of the mother and extrapolate the possible end-point. However, the result section strongly deserves the manuscript as it is confusing for the reader, and several interesting data are missing.
Throughout the manuscript, the authors write "40 pollen grains", "8 pollen grains", "17 pollen grains", and so on. I hope they would say "40 morpho-types of pollen grains", "8 morpho-types of pollen grains", "17 morpho-types of pollen grains", and so on. This syntactic formulation strongly deserves the result section and the discussion, as the reader really gets the impression that only 40 pollen grains were observed. The confusion is substantial, especially as the authors observed 700 individuals carrying pollen grains. Still, they provided no explanation in the Material and Methods section on how and why they selected and analysed only 40 pollen grains. I hope more than 400 pollen grains were analysed. Otherwise, the study would be very poor.
The result section strongly needs to be rewritten to fully present the size of the pollen data set and provide more information on the quantity of pollen grains analysed, the diversity of pollen grains per moth, the proportion between each morphotype, etc, etc, etc.
Moreover, the title mentions "regional pollination services", but the authors never question those services. They never question the viability of the pollen carried by the moths. They never questioned the way the moth visits the flowers. Can the moth's body parts carrying the pollen enter into contact with the stigma of the visited plants? Can the moths drop viable pollen on receptive stigmas of the same species? No data shows the pollination effectiveness of the moths.
So please make more efforts to present your interesting data more clearly. Try to be more careful about your conclusions on pollination services and discuss more the diversity of plants found on the moth´s bodies.
In fig 1 and 2 the authors present the "pollen carrier rate", but those data are not that interesting as they did not provide any data about the quantity of pollen by moth. An individual carrying a single pollen grain or thousands does not have the same interest for pollination purposes. Moreover, the authors did not provide any data on the diversity of pollen grains carried by moths. Later on, they focus on two plant species, but the reader needs to understand why they seem to be of particular importance. Are they two species dominant in the pollen carried by the moths?
L. 131 to 173. 700 individuals were recorded carrying pollen. The authors did not explain why they analysed only 40 pollen grains and did not explain how they chose these 40 pollen grains. Are they from a single individual? Are they from the same year? From the same month? Please specify why you studied so few pollen grains and how you selected them.
What is the pollen diversity per individual?
L.154 to 161. It is not clear how the authors constructed those data. They did not provide any details about which pollen grains were found on which individual moth. In the same way, it is very surprising that the 40 pollen grains analysed were all different.
If, but it is not what the text says, you identified 40 morpho-types of pollen grains, you will need to provide more data, i.e. about how many pollen grains of each type you identified on each individual moth, how many pollen grains of each type you identified per month, how many pollen types you identified per month, what the dominant pollen type is, etc. Based on such data, it will be possible to discuss what plants are the most visited by that moth species.
L. 198 areas
L 368 to 375. Which value of similarity did you use on BLAST to validate the identification?
L. 403 to 410. Where are the data analysed in that way?
L. 418-422 I think you overconclude. How long is the migratory journey? You did not analyse the viability of the pollen carried by the moths. There are no clues suggesting that the pollen is still viable after the migratory journey. The moths can play an essential role in pollination locally. However, pollination over a large geographical scale remains unshown, and you should only present it as a possibility that still needs to be investigated.
Author Response
The point-by-point responses to the comments from the reviewer 1
Q1: Throughout the manuscript the authors write "40 pollen grains" "8 pollen grains" "17 pollen grains", and so on I hope they would say "40 morpho-types of pollen grains" "8 morpho-types of pollen grains" "17 morpho-types of pollen grains" and so on. This syntactic formulation strongly deserves the result section and the discussion, as the reader really gets the impression that only 40 pollen grains were observed.
Response: Thank you very much for your professional and insightful comments. We agree that the wording in the manuscript might lead to misunderstanding. The expressions like "40 pollen grains" have been revised to "40 morpho-types of pollen grains" throughout the manuscript to emphasize the number of pollen types rather than the total pollen grains observed. These changes have been implemented to avoid confusion. Please refer to lines 138–146, 224, and 240–241.
Q2: The confusion is substantial especially as the authors observed 700 individuals carrying pollen grains, still they provided no explanation in the Material and Methods section on how and why they selected and analyzed only 40 pollen grains. I hope more than 400 pollen grains were analyzed. Otherwise, the study would be very poor.
Response: Revised. In our study, a total of 620 migratory S. litura individuals were found to carry pollen grains. Because pollen grains with extremely similar morphologies under the stereomicroscope were not entirely transferred to the SEM pin mount, consequently, we obtained 392 SEM images of pollen grains, which were categorized into 40 morpho-types based on their morphological characteristics. To address your concerns, we have added explanations about the selection process in the Materials and Methods section. Please see lines 372–373 and 379–380.
Q3: The result section strongly needs to be rewritten to fully present the size of the pollen data set and provide more information on the quantity of pollen grains analyzed the diversity of pollen grains per moth, the proportion between each morphotype etc, etc, etc.
Response: Accepted. We completely agree that more detailed quantitative analyses, such as the exact number of pollen grains per individual and the proportions between different morphotypes, would provide deeper insights into the ecological implications of our findings. However, due to the lack of suitable counting methods for quantifying pollen grains on individual moths with high accuracy, we focused instead on identifying the morpho-types and their relative proportions across the population. This approach allows us to explore the diversity of pollen grains carried by S. litura and their seasonal variations. To address your concerns, we have added a short paragraph and Table 3 to highlight the relative percentages of pollen families and their monthly differences. Please see lines 160–169 and Table 3.
Q4: Moreover, the title mentions "regional pollination services". but the authors never question those services. They never question the viability of the pollen carried by the moths. They never questioned the way the moth visits the flowers. Can the moth's body parts carrying the pollen enter into contact with the stigma of the visited plants? Can the moths drop viable pollen on receptive stigmas of the same species? No data shows the pollination effectiveness of the moths. So please make more efforts to present your interesting data more clearly. Try to be more careful about your conclusions on pollination services and discuss more the diversity of plants found on the moth's bodies.
Response: Revised. We fully agree that evaluating pollination services involves analyzing various factors, such as flower-visiting behavior, pollen viability, stigma receptivity, and subsequent seed set. Although our study did not directly examine the viability of pollen grains or the pollination behavior of moths, the discovery of pollen grains on migratory individuals of S. litura provides crucial evidence for its potential role in cross-regional pollination. As you pointed out, further studies are needed to confirm this possibility through detailed ecological and physiological analyses. To ensure more accurate and cautious expression, we have revised the title and related sections of the manuscript to emphasize the possibility of regional pollination services rather than making definitive conclusions. Additionally, we expanded the discussion on the diversity of pollen source plants found on the moths’ bodies. Please see lines 2, 27, and 272–282.
Q5: In Fig 1 and 2 the authors present the "pollen carrier rate", but those data are not that interesting as they did not provide any data about the quantity of pollen by moth. An individual carrying a single pollen grain or thousands does not have the same interest for pollination purposes. Moreover, the authors did not provide any data on the diversity of pollen grains carried by moths. Later on, they focus on two plant species, but the reader need to understand why they seem to be of particular importance. Are they two species dominant in the pollen carried by the moths?
Response: Revised. We calculated the pollen carrier rate to highlight the proportion of migratory individuals carrying pollen within the Spodoptera litura population, as this reflects the potential for pollination activity. We acknowledge that the ecological effects of carrying a single pollen grain versus thousands are different. However, due to the lack of suitable methods for accurately quantifying the exact number of pollen grains carried by each moth, our study focused on qualitative observations and identifying pollen morpho-types.
As for the two plant species you mentioned, we chose them because they are indicative of the source areas of S. litura migrants, which provides critical evidence for the moths' cross-regional migration activities. This was based on their geographical distribution and the ecological significance of their pollen. To address your concerns, we have added explanations in the text to clarify the importance of these two species. Please refer to lines 160–169 and 184–190. Additionally, while the diversity of pollen grains carried by moths was not analyzed on an individual level, we presented the relative proportions of different pollen families in Table 3 to provide a broader ecological context.
Q6: L.131 to 173 700 individuals were recorded carrying pollen. The authors did not explain why they analyzed only 40 pollen grains and did not explain how they chose these 40 pollen grains. Are they from a single individual? Are they from the same year? From the same month? Please specify why you studied so few pollen grains and how you selected them?
What is the pollen diversity per individual?.
Response: Revised. Similar concerns were addressed in Q2. We have clarified how we selected the 40 morpho-types of pollen for analysis in the Materials and Methods section. Please refer to lines 372–373 and 379–380.
Q7: L. 154 to 161 It is not clear how the authors constructed those data. They did not provide any details about which pollen grains were found on which individual moth. In the same way, it is very surprising that the 40 pollen grains analyzed were all different. If but it is not what the text says, you identified 40 morpho-types of pollen grains, you will need to provide more data, i.e. about how many pollen grains of each type you identified on each individual moth, how many pollen grains of each type you identified per month, how many pollen types you identified per month, what the dominant pollen type is, etc. Based on such data, it will be possible to discuss what plants are the most visited by that moth species.
Response: Accepted. As suggested, we have added a short paragraph and Table 3 to discuss the dominant pollen types and their distribution among individuals and months. Please see lines 160–169.
Q8: L.198 areas
Response: Accepted. The word “areaS” has been changed into “areas”. Please see line 215.
Q9: L. 368 to 375 Which value of similarity did you use on BLAST to validate the identification?
Response: Revised. In our study, we used the BLAST tool to align pollen DNA sequences against the reference database to identify pollen species. Specifically, we set the E-value threshold to ≤1e-5, which is a standard criterion to ensure high-confidence alignments. This threshold indicates a very low probability that the alignment occurred by random chance, thereby validating the accuracy of our identification. Additionally, we considered other key alignment metrics such as sequence identity (≥90%) and query coverage (≥90%) to further confirm the reliability of the matches. These criteria were chosen based on established practices in molecular identification to balance sensitivity and specificity. We have added this explanation to section “4.4.”
Q10: L.403 to 410 Where are the data analyzed in that way?
Response: Revised. The data analyzed in “Data Analysis” section were mainly described in “2.1”and ”2.2” of the “Results” section. Please see lines 404-410.
Q11: L.418-422 I think you overconclude. How long is the migratory journey? You did not analyze the viability of the pollen carried by the moths. There are no clues suggesting that the pollen is still viable after the migratory journey. The moths can play an essential role in pollination locally. However, pollination over a large geographical scale remains unshown, and you should only present it as a possibility that still needs to be investigated.
Response: Thank you very much for your professional and constructive comments. We agree that more cautious language is needed when discussing cross-regional pollination services. While our study did not directly assess pollen viability or pollination effectiveness, previous research using searchlight trap data (e.g., Fu et al., 2015) has clearly demonstrated the migratory behavior of S. litura, including its large-scale movements. In this study, we further applied insect source-tracing techniques, including trajectory analysis and pollen identification, to confirm that the tested individuals were migratory moths, which traveled across regions.
The discovery of diverse pollen types on these migratory individuals provides key evidence of their potential to transport pollen over large distances. Combined with their confirmed migratory range, our findings support the hypothesis that S. litura has the potential to participate in cross-regional pollination services. To address your concerns, we have revised the manuscript to present this as a possibility requiring further investigation, rather than a definitive conclusion. Please refer to lines 456–460 for the updated wording.
Reference:
Fu, X.W.; Zhao, X.C.; Xie, B.T.; Ali, A.; Wu, K.M. Seasonal pattern of Spodoptera litura (Lepidoptera: Noctuidae) migration across the Bohai Strait in northern China. Journal of Economic Entomology 2015, 108, 525-538. doi: 10.1093/jee/tov019
Reviewer 2 Report
Comments and Suggestions for Authors
Having done similar research myself, I am familiar with the methods and literature and am completely comfortable with this paper even if I do not have personal experience in China! I was somewhat surprised at the quantity and diversity of anemophilous pollen (Chenopods, Amaranths) borne by the moths. I have not observed that myself, but the pollen IDs appear fully credible. A valuable contribution to understanding seasonal Noctuid migration, which is a global phenomenon.
Author Response
The point-by-point responses to the comments from the reviewer 2
This reviewer did not provide specific revision suggestions for our manuscript.
Response: We greatly appreciate the recognition of our manuscript.
Reviewer 3 Report
Comments and Suggestions for Authors
The presented paper has solid science and my comments are primarily to improve the English and clarify a few points.
Page 1, Abstract
Line 15-change ecosystem to ecosystems
Line 22. Insert “of” between species plants
1. Introduction
Paragraph 5 (introduces study organism and impact), Recommend moving subordinate clause.
Previous studies, using various research methods…., have shown …..
Paragraph 6. Line 95. Change individual to individuals
2. Results
Page 3. Table 1. In legend. Specify if the plus/minus is standard deviation or standard error.
Page 3. Figure 1. Specify if the plus/minus is standard deviation or standard error.
Page 4. Figure 2. Specify if the plus/minus is standard deviation or standard error. Should the Figure 2 legend read P > 0.05 instead of P< 0.05 as not significant?
Page 4-It states that a total of 40 pollen grains were recorded? This is the number that were identified or the total number of pollen grains from all captured specimens? It seems really low if the latter. Or is a total of 40 different plant species which is what it appears to be in Table 2.
Page 4. Table. 2. Number 8. Missing the l in lavandulifolium
3. Discussion
Page 9, line 228, rephrase “research in respect of insect pollination networks” to “identify links in insect pollination networks”
Page 10, line 281. Not sure “airbags” is the best wording, recommend “wings” replacing airbags.
Lines 313-317. Clarify how much was estimated in the model used versus directly measured.
4. Materials and Methods
4.4 Word spacing is off on line 371 (check and revise)
Review of Supplemental Material
On Column 2, adjust width of the column so that Genus names are not split across two lines (rows)
On Column 3, the genus name is not needed as it is in Column 2, just include the plant family
On Column 4, remove the word plants as it is not needed, also herbaccous is misspelled, it should read herbaceous
On Column 5, substitute “flowering period” for “florescence”
On Column 6, add the word provinces to “Distribution range in China (provinces)” to clarify that these are provincial level distributions of the plants
Author Response
The point-by-point responses to the comments from the reviewer 3
Page 1, Abstract
Q1: Line 15-change ecosystem to ecosystems
Response: Revised. We have corrected the word. Please see the line 15.
Q2: Line 22. Insert “of” between species plants.
Response: Revised. We have inserted “of” between “species” and “plants”. Please see the line 22.
- Introduction
Q3: Paragraph 5 (introduces study organism and impact), Recommend moving subordinate clause.
Previous studies, using various research methods…., have shown …..
Response: Accepted. As suggested, we have adjusted the expression. Please see the lines 87-92.
Q4: Paragraph 6. Line 95. Change individual to individuals.
Response: Revised. We have corrected the word. Please see the line 95.
- Results
Q5: Page 3. Table 1. In legend. Specify if the plus/minus is standard deviation or standard error.
Response: Revised. The data in the last column were presented as “mean±SE”. We have changed the expression. Please see the line Table 1.
Q6: Page 3. Figure 1. Specify if the plus/minus is standard deviation or standard error.
Response: Revised. The mean data of pollen carrying rate was presented as “mean±SE.” Standard error (SE) was chosen as it provides a better representation of the variability of the mean for comparative purposes in this study. A sentence has been added to clarify this. Please see line 123.
Q7: Page 4. Figure 2. Specify if the plus/minus is standard deviation or standard error. Should the Figure 2 legend read P > 0.05 instead of P< 0.05 as not significant?
Response: Revised. The mean data was presented as “mean±SE”. “ns” indicates no significant difference base on the t-test (P>0.05). As suggested, we have corrected the expression. Please see the lines 133-135.
Q8: Page 4-It states that a total of 40 pollen grains were recorded? This is the number that were identified or the total number of pollen grains from all captured specimens? It seems really low if the latter. Or is a total of 40 different plant species which is what it appears to be in Table 2.
Response: Thanks a lot for your professional comments. For the numbers of “40 pollen grains” written in the manuscript, we would like to emphasize the number of pollen types rather than the number of pollen grains observed. Therefore, to avoid misunderstanding, the expression like "40 pollen grains" have been changed to "40 morpho-types of pollen grains" throughout the manuscript. Please see line 138-146, 224, 240-241.
Q9: Page 4. Table. 2. Number 8. Missing the l in lavandulifolium.
Response: Revised. We have corrected the word. Please see Table 2.
- Discussion
Q10: Page 9, line 228, rephrase “research in respect of insect pollination networks” to “identify links in insect pollination networks”
Response: As suggested, we have revised the short sentence. Please see the lines 245-246.
Q11: Page 10, line 281. Not sure “airbags” is the best wording, recommend “wings” replacing airbags
Response: Thank you for your suggestion. The term “airbags” was used to describe pollen grains with one or more air sacs, as observed in families such as Pinaceae and Podocarpaceae. These air sacs are formed by exinous expansions of the pollen wall, typically with an alveolate infratectum. To ensure clarity, we have replaced “airbags” with the more precise term “air sacs.” Please see line 309.
Q12: Lines 313-317. Clarify how much was estimated in the model used versus directly measured.
Response: Thank you for your advice. At this stage, we cannot directly monitor the flight parameters of high-altitude insects on Beihuangcheng Island. In the future, we will arrange a new type of high-resolution full-polarization insect radar to carry out insect migration monitoring. The radar can identify insect species, and then obtain the actual parameters of the migration of S. litura over the island. Therefore, we cannot compare the differences between the trajectory simulation results using these two types of parameters.
Q13: 4.4 Word spacing is off on line 371 (check and revise).
Response: As suggested, we have adjusted the word spacing. Please see the lines 403-404.
Review of Supplemental Material
Q14: On Column 2, adjust width of the column so that Genus names are not split across two lines (rows)
Response: As suggested, we have adjusted the width of the column. Please see the Table S1.
Q15: On Column 3, the genus name is not needed as it is in Column 2, just include the plant family
Response: As suggested, we have deleted the genus name in Column 3. Please see the Table S1.
Q16: On Column 4, remove the word plants as it is not needed, also herbaccous is misspelled, it should read herbaceous
Response: As suggested, we have deleted the word “plant” and corrected the word “herbaccous” to “herbaceous”. Please see the Table S1.
Q17: On Column 5, substitute “flowering period” for “florescence”
Response: As suggested, we have changed “florescence” to “flowering period”. Please see the Table S1.
Q18: On Column 6, add the word provinces to “Distribution range in China (provinces)” to clarify that these are provincial level distributions of the plants
Response: As suggested, we have added the word “provinces”. Please see the Table S1.
Reviewer 4 Report
Comments and Suggestions for Authors
This is a fine study that reports pollen occurrence on migratory Spodoptera litura. The methods are sound and combine classical morphological analysis with modern DNA techniques for identifying species of pollen origin. The figures and tables are appropriate and necessary providing readers a clear presentation of results and their import. I only noted a couple of minor corrections otherwise I think the manuscript is ready to proceed with publication.
On line 104 the appropriate verb is "was" not "were" ("A total...was"). In Table 1 and elsewhere the authors report results to the hundredth (0.01) but I think based on the size of their data sets the proper significant digit is to the tenth (0.1).
Author Response
The point-by-point responses to the comments from the reviewer 4
Q1: On line 104 the appropriate verb is "was" not "were" ("A total...was").
Response: Accepted. As suggested, we have corrected the verb. Please see the lines 104, 139, 237.
Q2: In Table 1 and elsewhere the authors report results to the hundredth (0.01) but I think based on the size of their datasets the proper significant digit is to the tenth (0.1).
Response: Thank you for your suggestion. Except for the statistical data on the proportion of trajectory landing points, we have made modifications according to your suggestions. Please see the lines 103-178.
Round 2
Reviewer 1 Report
Comments and Suggestions for Authors
The authors answered any comments and took advantage of them to improve the manuscript's value. They improved the reader's comprehension of their results and methods. They added some results and a table, enriching the value of their results and increasing the interest in their paper for ecologists.
I only have two remaining comments.
In line 400, the references for the pollen identification literature are missing.
For the DNA barcoding, I regret the authors provided a more precise answer in the report to the reviewer than in the manuscript… I think the thresholds used and some references could have been informative in the manuscript. In the response, the authors indicated a similarity threshold of 90% for species identification. 90% is low for an identification to the species level; it would only allow a confident identification at the genus level. See San Martin et al. 2024 (How reliable is metabarcoding for pollen identification? An evaluation of different taxonomic assignment strategies by cross-validation): "The identity score had also to be at least 97% for the species level, 90% for the genus level and 80% for the family level." However, we do not know which DNA sequence was used (was it the classical mitochondrial COI1?). Moreover, I guess that a lower identity, coupled with morphological clues, probably strongly compensates for the lower reliability of BLAST alignment. I think that such an explanation could make the Pollen Species Identification section more reliable.
Author Response
The point-by-point responses to the comments from the reviewer 1:
Q1: In line 400, the references for the pollen identification literature are missing.
Response: Thank you for pointing this out. Previously, we had provided references for each morphological type of pollen only in Table 2 of the “Results” section. Based on your suggestion, we have now added comprehensive references to the “Materials and Methods” section, specifically in line 397.
Q2: For the DNA barcoding, I regret the authors provided a more precise answer in the report to the reviewer than in the manuscript… I think the thresholds used and some references could have been informative in the manuscript. In the response, the authors indicated a similarity threshold of 90% for species identification. 90% is low for an identification to the species level; it would only allow a confident identification at the genus level. See San Martin et al. 2024 (How reliable is metabarcoding for pollen identification? An evaluation of different taxonomic assignment strategies by cross-validation): "The identity score had also to be at least 97% for the species level, 90% for the genus level and 80% for the family level." However, we do not know which DNA sequence was used (was it the classical mitochondrial COI1?). Moreover, I guess that a lower identity, coupled with morphological clues, probably strongly compensates for the lower reliability of BLAST alignment. I think that such an explanation could make the Pollen Species Identification section more reliable.
Response: As you rightly pointed out, a similarity threshold of 90% is insufficient for reliable species-level identification. We regret that our initial response may have caused confusion. To clarify, we did not intend to suggest that a 90% threshold should be used for species-level identification. Instead, our identification strategy adhered to a more nuanced approach, drawing on methodologies outlined in References 21 and 101. Specifically:
If the top five BLAST alignment results correspond to a single species, multiple species within a single genus, or multiple genera within a single family, the sequence was designated at the respective taxonomic level (species, genus, or family). Sequences aligning with multiple families were categorized as “unidentifiable.”
For species-level determination, we required a similarity threshold of 99% or higher, and for genus-level identification, a threshold of 95-99% was used. If the top five alignments included genera from different families, even at a high similarity threshold, the sequence was designated at the family level. For sequences with similarity below 90%, molecular identification was deemed unsuccessful.
Regarding San Martin et al. (2024), their use of ITS and rbcL barcodes for identifying plant species in bee bread provides a valuable reference. However, their study benefited from high-quality DNA templates extracted from pollen-rich samples. In contrast, our study faced unique challenges due to the nature of our samples. The migratory individuals of Spodoptera litura were collected in the field, and pollen grains were analyzed indoors after a considerable time interval. DNA templates were obtained from individual pollen grains exposed to high-energy electron beams under a scanning electron microscope, leading to inevitable DNA degradation.
In light of these constraints, we adopted a combined approach using both morphological characteristics and DNA barcoding for taxonomic identification. This integrated strategy ensured greater reliability in determining pollen source plants, even when the quality of the DNA template was suboptimal.
For clarity, the specific markers used in our study have been detailed in the “Materials and Methods” section (lines 390-391). These include rbcL and psbA-trnH from chloroplast DNA and ITS from nuclear DNA. As noted, the cytochrome c oxidase subunit I (COI) gene is commonly used for animal species identification and was not applicable in our study.